# Effect of Dietary *Laminaria digitata* with Carbohydrases on Broiler Production Performance and Meat Quality, Lipid Profile, and Mineral Composition

**DOI:** 10.3390/ani12081007

**Published:** 2022-04-13

**Authors:** Mónica M. Costa, José M. Pestana, Diogo Osório, Cristina M. Alfaia, Cátia F. Martins, Miguel Mourato, Sandra Gueifão, Andreia M. Rego, Inês Coelho, Diogo Coelho, José P. C. Lemos, Carlos M. G. A. Fontes, Madalena M. Lordelo, José A. M. Prates

**Affiliations:** 1CIISA—Centro de Investigação Interdisciplinar em Sanidade Animal, Faculdade de Medicina Veterinária, Universidade de Lisboa, 1300-477 Lisboa, Portugal; monicacosta@fmv.ulisboa.pt (M.M.C.); jpestana@fmv.ulisboa.pt (J.M.P.); cpmateus@fmv.ulisboa.pt (C.M.A.); catiamartins@isa.ulisboa.pt (C.F.M.); diogocoelho@fmv.ulisboa.pt (D.C.); jpclemos@fmv.ulisboa.pt (J.P.C.L.); carlos.fontes@nzytech.com (C.M.G.A.F.); 2Associate Laboratory for Animal and Veterinary Science (AL4AnimalS), Faculdade de Medicina Veterinária, Universidade de Lisboa, 1300-477 Lisboa, Portugal; 3LEAF—Linking Landscape, Environment, Agriculture and Food, Instituto Superior de Agronomia, Universidade de Lisboa, 1349-017 Lisboa, Portugal; diogo_osorio@outlook.pt (D.O.); mmourato@isa.ulisboa.pt (M.M.); mlordelo@isa.ulisboa.pt (M.M.L.); 4INSA—Departamento de Alimentação e Nutrição, Instituto Nacional de Saúde Doutor Ricardo Jorge, Avenida Padre Cruz, 1649-016 Lisboa, Portugal; sandra.gueifao@insa.min-saude.pt (S.G.); andreia.rego@insa.min-saude.pt (A.M.R.); ines.coelho@insa.min-saude.pt (I.C.); 5NZYTech—Genes and Enzymes, Estrada do Paço do Lumiar, Campus do Lumiar, Edifício E, 1649-038 Lisboa, Portugal

**Keywords:** *Laminaria digitata*, carbohydrate-active enzyme, animal growth, meat quality, poultry

## Abstract

**Simple Summary:**

Seaweeds represent promising alternatives to unsustainable conventional feed sources, such as cereals, incorporated in poultry diets. Brown macroalgae (e.g., *Laminaria digitata*) correspond to the largest cultured algal biomass worldwide and are rich in bioactive polysaccharides, minerals, and antioxidant pigments. However, their utilization as feed ingredients is limited due to the presence of an intricate gel-forming cell wall composed of indigestible carbohydrates, mainly alginate and fucose-containing sulfated polysaccharides. Therefore, supplementation with carbohydrate-active enzymes is required to disrupt the cell wall and allow seaweed nutrients to be digested and absorbed in poultry gut. The present study aimed to evaluate if the dietary inclusion of 15% *L. digitata*, supplemented or not with carbohydrases, could improve the nutritional value of poultry meat without impairing growth performance of broiler chickens. The results show that *L. digitata* increases antioxidant pigments and n-3 long-chain polyunsaturated fatty acids in meat, thus improving meat nutritional and health values. On the other hand, feeding algae at a high incorporation level impaired growth performance. Feed enzymatic supplementation had only residual effects, although alginate lyase decreased intestinal viscosity caused by dietary *L. digitata* with potential benefits for broiler digestibility.

**Abstract:**

We hypothesized that dietary inclusion of 15% *Laminaria digitata*, supplemented or not with carbohydrases, could improve the nutritional value of poultry meat without impairing animal growth performance. A total of 120 22-day old broilers were fed the following dietary treatments (*n* = 10) for 14 days: cereal-based diet (control); control diet with 15% *L. digitata* (LA); LA diet with 0.005% Rovabio^®^ Excel AP (LAR); LA diet with 0.01% alginate lyase (LAE). Final body weight was lower and feed conversion ratio higher with LA diet than with the control. The ileal viscosity increased with LA and LAR diets relative to control but without differences between LAE and control. The pH of thigh meat was higher, and the redness value of breast was lower with LA diet than with control. Meat overall acceptability was positively scored for all treatments. The γ-tocopherol decreased, whereas total chlorophylls and carotenoids increased in meat with alga diets relative to control. The percentage of n-3 polyunsaturated fatty acids (PUFA) and accumulation of bromine and iodine in meat increased with alga diets compared with control. Feeding 15% of *L. digitata* to broilers impaired growth performance but enhanced meat quality by increasing antioxidant pigments, with beneficial effects on n-3 PUFA and iodine.

## 1. Introduction

Macroalgae have been increasingly cultivated for numerous industrial applications, including biotechnological and nutritional purposes. Indeed, seaweeds are valuable sources of bioactive and prebiotic compounds (e.g., polysaccharides), minerals, vitamins (i.e., complex B, C, and E), pigments, essential amino acids, and bioactive peptides, with some species being good sources of polyunsaturated fatty acids [1].

Brown algae (*Phaeophyceae*), such as *Laminaria* sp., represent a large proportion of cultured seaweed biomass that can be used for feed and food [2]. *Laminaria* sp. is composed of bioactive polysaccharides (e.g., laminarin and fucoidan) with potential health benefits [3], as well as iodine and antioxidant carotenoids, chlorophylls, and vitamin E [1]. Although *Laminaria* sp. has low lipid content (up to 1.3% dry matter, DM), its fatty-acid profile may be rich in some polyunsaturated fatty acids (PUFA), such as arachidonic (20:4n-6, ARA) and eicosapentaenoic (EPA, 20:5n-3) acids [4,5], which are beneficial for human health [6]. In addition, dietary supplements of algal polysaccharide extracts were reported to modulate pigs’ gut microbiota, which can have an impact on lipid metabolism [7]. The nutritional and bioactive properties of *Laminaria* sp. were shown to enhance poultry meat quality [8,9,10], when these algae were used as feed supplements.

Several reports evaluated the potential of *Laminaria* sp. extracts as feed supplements for monogastric animals, mostly in the form of laminarin and fucoidan extracts [11,12]. However, to the best of our knowledge and despite the potential of using *Laminaria* sp. as a feed ingredient, such application was scarcely reported [13,14,15]. Indeed, high dietary levels of macroalga can compromise nutrient digestibility due to the presence of an intricate cell wall that is resistant to degradation by digestive enzymes, thus trapping other valuable nutrients and preventing their intestinal absorption [16]. In particular, brown seaweeds have a specific cell wall structure mainly composed of gel-forming alginate crosslinked with phenolic compounds and fucose-containing sulfated polysaccharides tightly linked with minor contents of cellulose [17]. Therefore, the use of exogenous carbohydrate-active enzymes (CAZymes) to degrade the brown macroalga cell wall is a promising strategy to increase the bioavailability of nutrients in poultry diets added with algae. Although there are some challenges related to large-scale and cost-effective algae production, the use of feed enzymes might allow seaweeds to be used as partial replacement sources of conventional and unsustainable feed ingredients (e.g., corn), enhancing the nutritional value of brown seaweeds by degrading algal non-starch polysaccharides [1]. This could be a solution to profit from the high biomass of macroalgae that can be produced per surface area and hinder the current food–feed–biofuel competition for conventional sources [1]. Commercially available CAZyme mixtures containing xylanases and β-glucanases have been widely incorporated in cereal-based diets for poultry to increase their nutritional value [18]. However, to date, there are no reports about the inclusion of exogenous enzymes in seaweed-added poultry diets. However, recent studies tested the benefits on growth and meat quality of using commercial (Rovabio^®^ Excel AP) and recombinant CAZymes as supplements in microalga-containing diets for broiler chickens [19,20]. In addition, alginate lyases and cellulases were shown to degrade *Laminaria digitata* biomass for biotechnological applications [21,22]. Moreover, in a recent in vitro study, an individual alginate lyase from a family 7 polysaccharide lyase (PL7) partially disrupted the *L. digitata* cell wall and released monounsaturated fatty acids, such as 18:1c9, and monosaccharides from algal biomass [23]. However, no in vivo assay was conducted in order to analyze the effect on broiler chicken growth and meat quality of supplementing recombinant alginate lyase in a diet incorporated with *L. digitata*. Therefore, the present study aimed to test if dietary supplementation with alginate lyase or a commercial carbohydrase would counteract the potential deleterious effects of adding high levels of *L. digitata* to the diet. Thus, feed enzymes are expected to improve the nutritional value of poultry meat by releasing algae bioactive compounds with benefits for human health. This would increase the importance and utilization of brown algae as a feed ingredient to partially replace corn.

## 2. Materials and Methods

### 2.1. Animal Management and Dietary Treatments

The procedures were approved by the Ethics Commission of CIISA/FMV and the Animal Care Committee of National Veterinary Authority (Direção Geral de Alimentação e Veterinária, Lisboa, Portugal), according to the guidelines of European Union legislation (2010/63/EU Directive). The experimental procedures with animals were also approved by ORBEA/ISA (protocol code number PTDC/CAL-ZOO/30238/2017, date of approval 7 July 2020).

A total of 120 1 day old male Ross 308 broiler chicks were housed in 40 wired-floor cages for 35 days, as previously described [19,20]. The initial body weight (day 0) of broilers was 44.7± 0.73 g and all animals were individually marked. Briefly, the birds were raised under environmentally controlled conditions, with continuously monitored temperature and ventilation. Three broilers were allocated to each pen with 10 replicate pens per treatment, in order to reduce the number of animals used in the experiment (3Rs principle) and according to previous studies [19,20,24], and subjected to an adaptation period of 21 days, where they were fed a corn- and soy-based diet. This was followed by an experimental period of 14 days, which corresponded to the broilers’ finishing period until the standard slaughter age of 35 days. During the experiment, birds received either a control diet with or without macroalgae incorporation or one of two enzyme supplementing treatments. The four experimental diets were as follows: (1) a corn–soybean-based diet (control); (2) the control diet with 15% *L. digitata* powder (Algolesko; Plobannalec-Lesconil, Brittany, France) (LA); (3) the LA diet supplemented with 0.005% commercial CAZyme mixture, Rovabio^®^ Excel AP (Adisseo; Antony, France) (LAR); (4) the LA diet supplemented with 0.01% recombinant CAZyme (LAE). Rovabio^®^ Excel AP is an enzyme complex with several synergistic carbohydrases but containing major activities for endo-1,4-β-xylanase (EC 3.2.1.8) and endo-1,3(4)-β-glucanase (EC 3.2.1.6) and other minor enzymatic side-activities. The enzymatic activity of β-xylanase is 3200 U/g, and that of β-glucanase is 4300 U/g. The recombinant CAZyme is an alginate lyase belonging to the PL7 family, which was shown to release 7.11 g/L of reducing sugars and 8.59 mmol/100 g dried *L. digitata* of monosaccharides [23]. Diets were finely ground and formulated to be isocaloric and isonitrogenous. The dietary ingredients are presented in Table 1.

During all trials, broilers were fed *ad libitum*, using a trough feeder, on a daily basis and had continuous access to water, without monitoring, through a nipple drinker. Animals and feeders were weighed once a week to obtain the average daily feed intake (ADFI), average daily gain (ADG), and feed conversion ratio.

At the end of the experiment, one broiler per pen was slaughtered by electrical stunning and exsanguination. Then, all gastrointestinal (GI) organs were manually removed, emptied using tap water, and weighed; the length was measured for the duodenum, jejunum, ileum, and cecum. The duodenum was collected between the end of the gizzard and of the pancreas; the jejunum was separated between the end of pancreas and Meckel’s diverticulum; the ileum was removed between Meckel’s diverticulum and the ileocecal junction; the cecum corresponded to the two large protuberations at the end of ileum.

The viscosity of small intestine contents was determined with a viscometer, as previously described [19]. Carcasses were air-chilled and monitored with a probe thermometer until an internal temperature of 4 °C. For the analysis of pigments, diterpenes, fatty acids, minerals, and lipid oxidative stability, breast (*pectoralis major*) and thigh muscles were removed from the left side of carcasses, minced, and stored at −20 °C. For determination of meat quality traits and sensory analysis, the procedures were conducted on the muscles from the right side of carcasses. However, for the sensory analysis and for iodine and bromine determinations, the breast was used instead of the thigh. This muscle selection was due to dissection difficulties of the thigh, in the case of the sensory analysis, and due to a higher representativeness (percentage of carcass weight) of breast than the thigh muscle, in the case of minerals. On the other hand, the thigh muscle was selected for determining lipid oxidative stability, since this muscle is more prone to oxidation due to an approximate twofold increase in lipid amount compared to breast muscle.

### 2.2. Production of Recombinant CAZyme

Plasmids containing the genes encoding the recombinant alginate lyase were obtained as described in a recent report [23]. Then, *Escherichia coli* (BL21) cells were transformed with the plasmids and were grown to mid exponential phase (absorbance between 0.4 and 0.6, λ = 595 nm) on Luria–Bertani medium at 37 °C, 200 rpm, with kanamycin (50 mg/mL). The recombinant gene was expressed in an NZY auto-induction LB medium (Nzytech, Lisbon, Portugal) incubated overnight at 25 °C, 140 rpm. Afterward, cells were submitted to ultrasonication and centrifugation, and the protein extract (supernatant) was recovered. Finally, the extract was freeze-dried and included, in equal weight proportions, at a final level of 0.01% in the LAE diet.

### 2.3. Chemical Analysis of L. digitata and Diets

The chemical composition of *L. digitata* and diets is presented in Table 2. The alga and feed DM, crude protein, ash, crude fat, and gross energy were determined using routine and widespread methods [20]. The metabolizable energy (ME) was calculated using the following formula: ME = 4412 − 11.06 × ash (g/kg DM) + 3.37 × crude fat (g/kg DM) − 5.18 × ADF (g/kg DM) [25]. The amino-acid composition of diets corresponds to estimated available proportions. Fatty acid methyl esters (FAME) of *L. digitata* and diets were obtained by one-step extraction and acidic transesterification [26] and analyzed using a gas chromatograph with a flame ionization detector (HP7890A Hewlett-Packard, Avondale, PA, USA) incorporated with a Supelcowax^®^ 10 capillary column (30 m × 0.20 mm internal diameter, 0.20 μm film thickness; Supelco, Bellefonte, PA, USA), following previously described conditions [20]. The internal standard was the nonadecanoic acid (19:0) methyl ester, and fatty acids were expressed as percentage of total fatty acids.

For the analysis of β-carotene and diterpenes (vitamin E homologs—tocopherols and tocotrienols), samples of *L. digitata* and diets (100 mg each) were weighed in duplicate, and the above compounds were extracted as reported by Prates et al. [27]. Samples were added with ascorbic acid followed by a saponification solution and were incubated and stirred in a water bath at 80 °C for 15 min. After saponification, n-hexane phases were separated by centrifugation (2500× *g* rpm, 10 min), filtered, and then analyzed in an HPLC system incorporated with a normal-phase silica column (Zorbax RX-Sil, 250 mm × 4.6 mm i.d., 5 μm particle size, Agilent Technologies Inc., Palo Alto, CA, USA) and two detectors set in series, according to conditions previously described [19,20]. The compounds were determined following the external standard technique and using a standard curve of peak area versus concentration.

The analysis of pigments of *L. digitata* and diets was conducted according to Teimouri et al. [28] but with some modifications reported by Pestana et al. [19]. Briefly, 0.5 g of samples were stirred with 5 mL of acetone in the dark. The homogenized sample mixture was centrifuged (3000× *g* rpm, 5 min, 4 °C) and the supernatant was separated. Chlorophyll a and b and total carotenoids were detected at 645 and 662 nm and at 470 nm, respectively, using UV/Vis spectrophotometry (Ultrospec 3100 pro, Amersham Biosciences, Little Chalfont, UK). The concentrations of pigments were calculated using the equations reported by Hynstova et al. [29].

The mineral profiles of *L. digitata* and diets were determined according to Ribeiro et al. [30]. Briefly, 0.3 g of lyophilized samples were weighed in a digestion tube and added with concentrated nitric acid (3 mL) and concentrated hydrochloric acid (10 mL). Then, samples were incubated in a ventilated chamber for 16 h followed by the addition of 1 mL of hydrogen peroxide. Afterward, the samples were heated in a digestion plate (DigiPREP MS, SCP Science, Baie-D’Urfé, QC, Canada) as follows: 1 h to reach 95 °C and 1 h at 95 °C. After digestion, samples were left to cool and then diluted with distilled water for a final volume of 25 mL and filtered through filter papers (90 mm diameter) into sealed flasks. The samples were analyzed for the different elements by inductively coupled plasma optical emission spectrometry (ICP-OES, iCAP 7200 duo Thermo Scientific, Waltham, MA, USA). Multi-element standards (PlasmaQual S22, SPC Science, Baie-D’Urfé, QC, Canada) were used to create the calibration curves necessary to quantify the different elements (Ca, K, Mg, Na, P, S, Cu, Fe, Mn, Zn, Cr, Cd, Ba, V, Ni, Pb, Co, and As).

The determination of iodine and bromine was carried out using an inductively coupled plasma mass spectrometer (ICP-MS) (Thermo X series II, Thermo Fisher Scientific, Waltham, MA, USA) preceded by an alkaline extraction, according to Delgado et al. [31]. Succinctly, approximately 0.2 g of freeze-dried *L. digitata* and diet samples were weighed into a 50 mL tube, followed by the addition of tetramethylammonium hydroxide (TMAH) solution at 25% v/v (1 mL) and ultrapure water (8 mL) (Milli-Q Element system, Millipore Corporation, Saint-Quentin, France). The samples were extracted in triplicate and spiked with chemical standards to ensure the analytical quality, in a Heating Graphite Block System (DigiPREP MS, SCP Science, Baie-D’Urfé, QC, Canada) for 3 h at 90 °C. After extraction, samples were centrifuged and filtered with 90 mm filters (Filtros Anoia S.A., Barcelona, Spain).

### 2.4. Evaluation of Meat Quality Traits

The procedures for determination of meat pH, color, shear force, and cooking loss were previously reported [19,20]. Briefly, pH was measured 24 h postmortem, in triplicate, on skinless and deboned muscles with a glass penetration pH electrode (HI9025, Hanna instruments, Woonsocket, RI, USA). The color parameters (CIELAB; lightness (L*), redness (a*), and yellowness (b*)) were measured in triplicate on muscles using a Minolta CR-300 Chromameter (Minolta camera Co. Ltd., Osaka, Japan), after the carcass was cooled for 24 h and the meat was exposed to air for 1 h. Then, muscles were stored in vacuum-sealed plastic bags at −20 °C until cooking loss and shear force analyses. Afterward, meat was thawed at 4 °C for 24 h, cooked at 80 °C to reach a monitored internal temperature of 72 °C, and kept at room temperature for 2 h. Muscles were weighed before and after cooking for cooking loss determination. Afterward, muscles were cut into strips (1 cm × 1 cm × 5 cm), and shear force was measured using a texture analyser TA.XTplus (Stable Microsystems, Surrey, UK) incorporated with a Warner–Bratzler blade and expressed as the mean peak value of at least four replicates.

### 2.5. Sensory Analysis by a Trained Panel

The sensory analysis was conducted according to the procedures described by Pestana et al. [19]. Briefly, meat samples were cooked at 80 °C in plastic bags until a monitored internal temperature of 78 °C. Then, they were cut (about 1 cm^3^), and eight randomly selected samples per plaque were maintained at 60 °C for each of the five panel sessions. The trained sensory panel was composed of 10 trained panelists from the Faculty of Veterinary Medicine (University of Lisbon, Lisbon, Portugal). Four attributes were evaluated (tenderness, juiciness, flavor, off-flavors, and overall acceptability) by the panelists and classified according to an eight-point scale (1 being extremely tough, dry, weak, and negative; 8 being extremely tender, juicy, strong, and positive), except flavor and off-flavor that were scored from 0 (absence) to 8 (very intense).

### 2.6. Evaluation of Total Cholesterol, Diterpenes, Pigments, and Minerals in Meat

Total cholesterol, and homologs of vitamin E were extracted, in duplicate, from fresh muscles (750 mg each), following the same procedure described for alga and diets, using direct saponification, single *n*-hexane extraction, and HPLC analysis [19,27].

For determination of pigments in the muscles, 2.5 g of breast and thigh were weighed and stirred with 5 mL of acetone in the dark. This mixture was homogenized with a homogenizer (Ultra-Turrax T25, IKA-Werke GmbH&Co. KG, Staufen, Germany) for 1 min, and then centrifuged at 3000× *g* rpm for 5 min. The supernatant was measured at the same wavelengths described for macroalga and diets. The pigment contents were determined according to Hynstova et al. [29].

The mineral profiles were determined in freeze-dried muscle, following the same procedure as *L. digitata* and diet samples, except for the amount of sample weighed (around 0.6 g) for iodine and bromine analyses.

### 2.7. Determination of Lipid Oxidative Stability in Meat

Lipid peroxidation in meat was determined by measuring thiobarbituric acid reactive substances (TBARS) concentration, following the spectrophotometric method described by Grau et al. [32], after four minced portions of meat (1.5 g each) were stored for 0 and 6 days at 4 °C. The procedure was conducted as previously described by Martins et al. [33]. The presence of a pink chromogen derived from TBARS (e.g., malonaldehyde, MDA) was analyzed by measuring the absorbance at λ = 532 nm using a UV/visible spectrophotometer (Ultrospec III, Pharmacia LKB Biochrom Ltd., Cambridge, UK). For TBARS quantification, the precursor of malonaldehyde 1,1,3,3-tetraethoxypropane (Fluka, Neu Ulm, Germany) was used to build the standard calibration curve, and the results were expressed as mg of MDA/kg of meat.

### 2.8. Determination of Total Lipids and Fatty-Acid Composition in Meat

The extraction of total lipids from freeze-dried muscle samples was conducted, in duplicate, using dichloromethane–methanol (2:1, v/v), and lipids were determined gravimetrically [34].

The fatty acids were transesterified into FAMEs using a combination of basic followed by acidic catalysis [19]. Then, FAMEs were separated in a Supelcowax^®^ 10 capillary column (Supelco, Bellefonte, PA, USA) using gas chromatography with flame ionization detection, and the running conditions were as previously described [20]. Fatty acids were identified by comparison with a standard (FAME mix 37 components, Supelco Inc., Bellefonte, PA, USA), quantified using 19:0 methyl ester as an internal standard, and expressed as percentage of total fatty acids.

### 2.9. Statistical Analysis

ANOVA from GLM of the Statistical Analysis System (SAS) program (SAS Institute Inc., Cary, NC, USA) and the adjusted Tukey–Kramer method (PDIFF option) for multiple comparisons of least squares means were used for data analysis. The PROC POWER model of SAS was applied for evaluation of statistical power. The experimental unit was either the cage for ADFI and feed conversion ratio or the animal for body weight (BW), ADG, and meat quality variables. The treatment was considered a fixed factor in the model. The statistical significance was assumed as α *=* 0.05. A principal component analysis (PCA) was conducted with muscle chemical parameters using the Statistica program (version 8.0; TIBCO software, Palo Alto, CA, USA).

## 3. Results

### 3.1. Growth Performance of Chicken and Digestive Tract Parameters

The effect of dietary treatments on the growth performance and gastrointestinal tract of broiler chickens is presented in Table 3. The final average BW of birds consuming the LA and LAE diets was 185 g lower (*p* = 0.011) than the control birds. Final BW did not differ (*p* > 0.05) between broilers fed the LAR diet and the control. The feed conversion ratio of birds fed the LA treatment was higher (*p* = 0.012) than those fed a control diet. The ADG was 1.17-fold lower (*p* = 0.039) for broilers fed the LA diet than for those fed the control, but without differences (*p* > 0.05) between LAR and LAE treatments and the control. The mortality of broiler chickens was low (2.5%) (data not shown), since, during the experimental period, only two animals from LAR and one from LAE treatments presented diarrhea followed by death. A decrease in gizzard weight (*p* = 0.013) was found in the animals fed macroalga-containing diets compared to the control. Conversely, ileum and cecum weights were higher (*p* < 0.05) in broilers fed LA and LAR treatments, respectively, in comparison with broilers fed the control diet. No differences were found in birds’ ileum and cecum weights between LAE and control treatments. In addition, jejunum and ileum lengths increased (*p* < 0.05) in broiler chickens fed macroalgae-added diets compared with those fed the control. Similarly, the cecum length was higher (*p* = 0.001) with LA and LAE treatments than with the control, but no differences were found between LAR and control. The viscosity of ileum contents of birds fed LA or LAR diets was higher (*p* < 0.001) than the control, but without differences (*p* > 0.05) between the LAE group and control.

### 3.2. Meat Quality Traits and Sensory Analysis

The influence of treatments on the meat quality traits of chickens is presented in Table 4. The pH of breast muscle, 24 h postmortem, was lower (*p* = 0.020) with the LAE than with the LA treatment, but without differences (*p* > 0.05) between macroalga-containing treatments and control. However, the pH of thigh muscle was higher (*p* = 0.035), whereas the redness (a*) value of breast was lower (*p* = 0.032) with LA treatment than with the control. Cooking loss and shear force were not affected by treatments (*p* > 0.05) in breast and thigh muscles. The sensory analysis of breast meat is shown in Table 5. The meat juiciness was lower (*p* = 0.002) with LAE treatment than with LAR and control. Meat flavor was more intense (*p* = 0.007) with the control than with LAE treatment. Off-flavor and overall acceptability were not affected (*p* > 0.05) by treatments.

### 3.3. Diterpene Profile, Pigments, and Oxidative Stability

Vitamin E and pigment profiles in breast and thigh muscles are presented in Table 6. In the breast, the γ-tocopherol amount was lower (*p* < 0.001) with the macroalga-containing treatments (0.052 µg/g) than with the control (0.072 µg/g), whereas, in the thigh, this compound was only significantly reduced (*p* = 0.030) with the LAE treatment. The total chlorophylls (*p* < 0.001) and carotenoids (*p* = 0.002) were increased by the treatments with macroalga in relation to control, in both muscles. In fact, highly significant (*p* < 0.001) increments in chlorophyll a and b were found in meat due to the presence of *L. digitata*, with an almost twofold increase in total chlorophylls in the breast muscle with the macroalga-containing treatments compared with the control. In addition, the dietary treatments had no significant effect (*p* = 0.111) on the oxidative stability of thigh meat after 6 days of storage at 4 °C (Table 7), although TBARS values were numerically lower in the meat of broilers fed macroalga diets (average of 0.346 mg/kg) than with the control (0.702 mg/kg).

### 3.4. Total Lipids and Fatty-Acid Profile

The influence of dietary treatments on total lipids and fatty-acid profiles of breast and thigh muscles is shown in Table 7 and Table 8, respectively. The treatments with macroalgae led to a 1.4-fold decrease (*p* < 0.01) of total lipids compared to control, in both muscles. The presence of *L. digitata* also changed muscle fatty-acid composition. The percentages of 14:0 (*p* < 0.01) and 16:0 (*p* < 0.001) decreased with the macroalga-containing treatments in thigh and breast muscles, but no differences (*p* > 0.05) were found for 14:0 proportion in the breast between LAR treatment and control. In addition, in the breast muscle, the percentages of 14:1c9 (*p* = 0.006) and 16:1c7 (*p* < 0.001) were lower with the macroalga diets than with the control. In the thigh muscle, these monounsaturated fatty acids (MUFAs) were also decreased (*p* < 0.05) with the LA and LAR when compared with the control. The proportion of 16:1c9 was lower with LA and LAE treatments in the breast (*p* = 0.014), and with the LA treatment in the thigh (*p* = 0.033), in relation to the control. The 18:1c11 MUFA was the only one increased (*p* < 0.05) by the dietary incorporation of macroalgae, and these differences were highly significant (*p* = 0.001) in the thigh muscle. Considering the percentage of PUFAs, a decrease in 18:2t9,t12 was found with LAR and LAE treatments in the breast muscle (*p* = 0.027) and with LA and LAR treatments in the thigh muscle (*p* = 0.003). Similarly, the proportion of 18:3n-3 was lower (*p* < 0.01) with the macroalga diets than with the control, in both muscles. However, 18:3n-6 (*p* = 0.001) was increased with LAR and LAE in the breast. The percentages of 20:1c11 and 20:3n-6 were decreased (*p* = 0.001) with the macroalga-containing treatments relative to the control, in breast and thigh muscles, respectively. The proportions of 20:3n-6 in the breast and 20:3n-3 in the thigh were significantly decreased (*p* < 0.05) with the LAR treatment when compared to the control. Conversely, an increase (*p* = 0.006) in 20:4n-6 was found in the thigh muscle with the LA treatment in relation to control. In addition, the percentages of 20:5n-3 and 22:5n-3 were highly significantly increased (*p* < 0.001) in both muscles with the macroalga diets when compared with the control. Similar results were found for the 22:6n-3 proportion (*p* ≤ 0.001), although this fatty acid was not increased (*p* > 0.05) with the LAR treatment in the breast muscle.

The total of *cis*-MUFA in the breast was slightly but significantly (*p* = 0.045) decreased with the LA treatment (25.8%) compared with the control (28.4%). Conversely, the sum of PUFA in the breast was higher (*p* < 0.001) with the macroalga-containing treatments than with the control. This difference was due to a 1.06-fold increase in n-6 PUFA (*p* = 0.003) and a 1.7-fold increment in n-3 PUFA (*p* < 0.001). Similar but less significant (*p* = 0.041) results were found in the thigh muscle for the total of PUFA, with a tendency (*p* = 0.053) for a higher percentage of n-6 PUFA with the macroalga diets and a highly significant (*p* < 0.001) increase in n-3 PUFA with LA and LAE treatments compared with the control. Therefore, an increased (*p* < 0.05) PUFA/SFA ratio was found with the macroalga containing treatments, in the breast, and with the LAE treatment, in the thigh, in relation to the control. In addition, the n-6/n-3 PUFA ratio decreased (*p* < 0.001) 1.5-fold in the breast and 1.2-fold in the thigh with the macroalga diets in comparison with the control.

### 3.5. Mineral Composition

The effect of dietary treatments on the mineral composition of breast and thigh meats is presented in Table 9. The treatments had no impact (*p* < 0.05) on the individual or total amount of macrominerals, in both muscles. However, in the breast muscle, highly significant increases (*p* < 0.001) in bromine and iodine of 0.67 and 0.34 mg per 100 g of meat, respectively, were found with the macroalga diets in relation to control, which led to a 1.3-fold increment (*p* < 0.001) in total microminerals.

### 3.6. Principal Component Analysis

The PCA, relating the different parameters of muscle chemical composition with the four dietary treatments, is presented in Figure 1, for the breast, and in Appendix A, for the thigh. Considering the breast muscle, the two-dimensional variability of pooled data in Figure 1a,b shows a good separation between the treatments with macroalga and the control. Indeed, data from LA, LAE, and LAR treatments are aggregated in quadrants a, c, and d, whereas data from the control are accumulated in quadrant b. The total data variability was explained at approximately 43.7% by the first two discriminant factors (29.6% for factor 1 and 14.1% for factor 2). The loadings for the two factors are presented in Table 10. The variables with the highest discriminant power were 14:0, 18:1c9, 18:3n-3, 20:4n-6, 22:5n-3, and 22:6n-3, for factor 1, and 16:0, 18:2n-6, 20:3n-6, and iodine, for factor 2. For the thigh muscle, a separation between macroalga-containing treatments and the control was also obtained with the PCA model. However, without the presence of iodine and bromine as variables, this distinction was not as clear as for the breast muscle since a greater data dispersion was found in the thigh, as shown in Appendix A. Data from LA, LAE, and LAR treatments were mostly aggregated in quadrants a and c, but some points were present in quadrants b and d together with data from the control. The two principal components explained about 37.6% of total data variability, with a lower contribution of factor 1 (23.0%) and a similar contribution of factor 2 (14.6%) compared with that obtained for breast muscle. The highest discriminant variables were 14:0, 20:4n-6, 22:5n-3, and 22:6n-3, for factor 1, and 16:0, 18:0, 18:2n-6, 18:3n-3, and 22:1n-9, for factor 2 (Appendix A).

## 4. Discussion

The dietary inclusion of 15% *L. digitata* as a partial substitute of corn impaired animal growth performance. Indeed, reductions of 11% in final BW and 14% in ADG, along with a consequent increase of 10% in feed conversion ratio, were found in birds fed the macroalga treatment without the addition of enzymes. Although there was a numerical difference between the initial BW (day 21) of broilers from the control and LA treatment due to a considerable variability of values, it was not significant and, thus, this parameter cannot be considered a conditioning factor of final BW (day 35). In addition, the difference in BW at day 35 (192 g) between control and LA treatment was almost four times higher than that found at day 21 (49.9 g), which was attributed to the dietary incorporation of algae. The commercial CAZyme mixture with major enzymatic activities of xylanase and β-glucanase slightly counterbalanced the negative effects on BW and ADG caused by LA treatment, which was also found with the recombinant alginate lyase for ADG and feed conversion ratio. The negative effects on chicken growth caused by dietary *L. digitata* were probably due to the high inclusion rate of alga, since low doses of seaweed extract were described to enhance growth performance of broilers [11,12] and pigs [35,36,37,38,39,40]. For instance, final BW was increased in broiler chickens fed *Laminaria* spp. extract supplemented at 20 mg/mL in water [10], and feed efficiency was enhanced in pigs fed with up to 0.03% of *Laminaria* sp. extracted polysaccharides (e.g., laminarin and fucoidan) [35,37,39]. Conversely, Ventura et al. [15] found that the incorporation of *U. rigida*, at more than 10% in a broiler starter diet, increased feed conversion ratio, which was suggested to be due to the presence of high amount of indigestible algal polysaccharides. In addition, Zahid et al. [41] showed that a brown alga mixture led to a numerical decrease in final BW when fed at 20% to 40% to chicks. Recently, Stokvis et al. [14] described an increase in feed conversion ratio in broilers fed 10% of the brown macroalga, *Saccharina latissima*. The latter results were previously described by Bikker et al. [42] as being caused by the high mineral and non-starch polysaccharide contents of brown macroalgae. In the present study, there were considerable high levels of minerals in the seaweed-containing diets, but broilers’ mortality was less than 3%, with few animals presenting diarrhea that compromised their growth and health, conversely to what was suggested by Bikker et al. [42]. Therefore, the presence of algal indigestible polysaccharide in the diets was probably the cause of animal growth impairment. Despite the fact that nutrient digestibility was not evaluated, the increase by more than 40% in ileal viscosity in chickens fed LA and LAR diets in comparison with the control and LAE might have reduced feed passage and consequently nutrient digestion and absorption by trapping valuable nutrients [43]. The increment in ileal viscosity caused by the dietary incorporation of *L. digitata* was probably due to the presence of hydrocolloidal and anionic polysaccharides in the macroalgal cell wall (e.g., alginates) [44] that are largely indigestible by monogastric animals and can increase medium viscosity [45]. Similar effects were previously reported when 15% or 10% of the microalgae *Arthrospira platensis* [19] and *Chlorella vulgaris* [20], respectively, were fed to broiler chicks. However, contrary to what was described for microalga, the recombinant enzyme reversed the effect on ileal viscosity caused by the macroalga treatment. Therefore, it is possible that alginate lyase partially degraded the *L. digitata* cell wall, as previously shown in vitro [23], and, to some extent, disrupted gelling and hydroscopic polymers formed by crosslinking between cell-wall phenolic compounds and alginate [17]. In addition, the xylanases and β-glucanases in LAR treatment could not specifically hydrolyze algal polysaccharides, and, as expected, an increase in intestinal viscosity was also observed when the LA diet was supplemented with the commercial carbohydrase mixture.

Furthermore, in the present study, the increase in jejunum and ileum lengths with all macroalga-containing treatments and of cecum length with LA and LAE treatments could indicate a morphological intestinal modification to compensate growth impairment and, thus, increase nutrient absorption. However, it is possible that this effect was not just a compensatory mechanism but also the result of algal polysaccharide bioactivity. In fact, laminarin and fucoidan extracted from *Laminaria* sp. were previously shown to increase duodenal villous height in piglets [46], even though their effect on intestinal length was not reported.

Considering meat quality, the pH of thigh meat was significantly, although slightly, increased by the LA treatment relative to control. The pH was shown to be a determinant factor of myofibrillar protein denaturation in red muscles [47] with a potential effect on meat sensory attributes and carcass traits [48]. However, pH values in the thigh, similarly to those found in the breast, were within the range previously described for poultry meat [10,49] and, thus, this parameter was not be associated with any modification of meat quality. Moreover, the color of breast meat was influenced by dietary macroalga, with a decrease in a* value promoted by the LA treatment. A similar result was reported by Tavaniello et al. [10] in chicken breast muscle with in-water *Laminaria* spp. supplementation, and by Rajauria et al. [50] in pork with a low dietary level (0.53% feed) of *Laminaria* spp. extract. In the present study, the analytical difference in meat color was not distinguishable by the naked eye in the raw meat or even by the trained sensory panel exposed to cooked meat. Therefore, this modification would not have a negative impact on the appearance of meat for consumers. The effect of macroalga treatments on meat redness might be explained by the 1.4-fold increase in total carotenoids in breast and thigh muscles. In addition, meat color was probably determined by myoglobin concentration and oxidation status in the muscle, since oxidation of myoglobin into oxymyoglobin is responsible for the bright cherry-red color sought by consumers [51]. In fact, an interaction between positively charged proteins in meat, such as oxymyoglobin, and algal anionic polysaccharides may have occurred with a consequent effect on meat redness. This aspect was suggested by Moroney et al. [52] to justify a decrease in a* value, in a concentration-dependent manner, in pig meat pulverized with *L. digitata* extract containing fucoidan and laminarin. The fact that the use of alginate lyase could counterbalance the effect of macroalga on meat redness corroborates the phenomenon described by Moroney et al. [52]. Interestingly, the dietary supplementation with commercial CAZyme mixture led to a similar result to that found with the LAE treatment, although minor degradation activities toward *L. digitata* polysaccharides were previously reported for xylanases and β-glucanases [23].

The increase in total carotenoids and chlorophylls in chicken breast and thigh meats provides benefits for consumers and enhances the nutritional value of meat. For instance, fucoxanthin, which is the major carotenoid present in brown seaweeds, was shown to have antioxidant, antitumor, and anti-inflammatory properties [53]. Although chlorophyll metabolism and function have been scarcely studied, Viera et al. [54] reported that, in mice, chlorophylls are converted into pheophorbides or pheophytins, absorbed in the intestine, and eventually transported to tissues. This process of conversion of chlorophylls into their derivatives and uptake by cells was also demonstrated in vitro with seaweeds, such as *Laminaria ochroleuca* [55,56]. Chlorophylls and their derivatives were shown to have important functional functions, such as the ability to trap mutagens and antioxidant activities. The latter activities include free-radical-scavenging properties and metabolic activation of detoxification pathways [57]. As a matter of fact, chlorophylls, such as chlorophyll *a*, were identified as antioxidants having a synergistic activity with vitamin E because of their ability to scavenge peroxyl radicals [58]. Regarding vitamin E homologs, the treatments with macroalga had no effect on the amount of the major homolog α-tocopherol, which was within the range described for broiler chicks [19,59]. However, γ-tocopherol was decreased with macroalga-containing treatments in the breast and with LAE treatment in the thigh. The fact that a considerably low amount of γ-tocopherol (up to 0.075 µg/g) was obtained in both meats indicates the reduced biological impact of these results. In addition, the concentration of γ-tocopherol was about 10 times lower than that found in previous studies [19,20,59], which was mostly due to the decrease in γ-tocopherol amount, between 38.5% and 72.3%, in the present control diet compared with previous diets. Moreover, the sensory panel detected a decrease in juiciness and flavor of breast meat with LAE treatment compared with the control. The mechanisms responsible for the changes in the panel perception of meat juiciness and flavor remain to be explained. However, although these parameters were significantly discriminated, they changed by less than 1.0 and, thus, no major impacts on consumer acceptability of meat are expected. Indeed, meats from all treatments were positively (>4.0 points) scored for overall acceptability without differences for this parameter.

The dietary treatments influenced the total lipid amount and the fatty-acid profile of breast and thigh. In fact, treatments with *L. digitata* led to a reduction in total lipids in both meats and, thus, to an increase in meat leanness, which was described as one of the major attributes determining consumers’ decisions toward meat [60].

Nevertheless, meats from all treatments were considered lean (total fat < 5%) [61], with an average of total lipids of 1.8% for the breast and 2.6% for the thigh. In addition, treatments with macroalgae promoted the accumulation of total PUFA with increases in n-6 and n-3 PUFAs in the breast and in n-3 PUFAs in the thigh, although the latter was not significant with the commercial CAZyme supplementation. The higher accumulation of n-3 PUFA in meat with alga-added diets was mainly due to a relative increment in specific n-3 long-chain (LC) PUFAs, including 20:5n-3, 22:5n-3, and 22:6n-3. Similar findings were previously reported when *Laminaria* spp. were used as a supplement for broiler chickens, except for the effect on 20:5n-3 [10]. Furthermore, Islam et al. [9] described that dietary supplementation with a mixture (1:1) of *Laminaria japonica* and charcoal at up to 1% led to an increase in 22:6n-3 proportion in duck meat. Considering the low lipid content (1.31% DM) of *L. digitata*, it is possible that the beneficial contribution of this macroalga for n-3 LC PUFA proportion in meat was not just a result of the high percentage of these fatty acids in seaweed biomass, but also a consequence of the bioactivity of algal polysaccharides. In fact, previous reports in pigs showed that polysaccharides extracted from *Laminaria* spp. (i.e., fucoidan and laminarin) can modify gut microbiota and, consequently, the production of short-chain fatty acids [7]. The latter were shown to be involved in lipid metabolism and de novo synthesis of fatty acids in the liver [62]. Although the effect of algal polysaccharides on the gut microbiota and lipid metabolism was not assessed in the present study, the high levels of *Laminaria* spp. incorporated in the broiler diet could have potentiated the action of polysaccharides toward fatty-acid metabolism. The relative increase in n-6 and n-3 PUFAs is particularly relevant, considering the low efficiency of conversion pathways of 18:3n-3 and 18:2n-6 into n-3 LC-PUFA and n-6 LC-PUFA, respectively, and the reduced human intake of PUFA, mostly n-3 PUFA [63]. The importance of enriching chicken meat with n-3 LC-PUFA is linked to the benefits of these fatty acids for human health, since n-3 LC-PUFAs are associated with enhanced cognitive abilities and suppression of chronic diseases, such as rheumatoid arthritis, atherosclerosis, and coronary heart disease [6,63].

Herein, the effect of treatments on individual fatty-acid proportions led to an increase in PUFA/SFA ratio in the breast and a decrease in n-6/n-3 PUFA ratio in breast and thigh with macroalga-containing treatments. Therefore, the dietary incorporation of *L. digitata* improved meat nutritional value, since a higher PUFA/SFA ratio and lower n-6/n-3 PUFA ratio have been used as indicators of healthier meat [64]. However, the n-6/n-3 PUFA ratio was considerably above the maximum recommended value of 4.0 [64], which was essentially due to a predominance of 18:2n-6 (more than 30%) among all fatty acids present in meat.

Moreover, the treatments with macroalga led to a significant increase in iodine and bromine in the breast muscle of chickens and, consequently, to an increment in total microminerals in meat. A similar result for iodine was found in the adipose tissue and muscle of piglets fed a diet supplemented with 0.116% or 0.186% *L. digitata* [65]. In the present study, the accumulation of iodine and bromine in meat was explained by a considerable increase in each one of these compounds in alga-added diets compared with the control diet. In fact, *L. digitata* presented high levels of iodine (4399 mg/kg DM) and bromine (474 mg/kg DM), which are within the range of values already reported, due to the ability of seaweed to concentrate mineral compounds from seawater [1]. The increase in iodine in poultry meat can be favorable for human health, particularly considering the low iodine intake of nearly one-third of the global population, mainly children and pregnant women, with a consequent impairment of thyroid hormone synthesis [66,67]. The lack of thyroid hormones compromises cellular metabolism and development of organs, especially the brain, and eventually leads to iodine deficiency disorders [67]. Although a higher accumulation of iodine in meat can improve its nutritional value, an excessive intake of this mineral should be monitored to avoid pathological problems. In the present study, meat from treatments with macroalgae contained an average of 0.35 mg of iodine/100 g, which is above the recommended daily intake of 0.15 mg/day for an adult person [67] ingesting 100 g of meat per day. Therefore, a meat with an accumulation of iodine higher than that with the control (0.01 mg/100 g) but lower than with macroalgae would be more favorable. In addition, bromine has no proven nutritional benefits, and it is considered a food contaminant and a potentially toxic element [68]. However, the amount of bromine present in the meat from treatments with macroalgae (average of 790 µg/100 g; 11.3 µg/kg BW/d for 70 kg of BW) is far below the acceptable daily intake of 1000 µg/kg BW/day [68].

## 5. Conclusions

The dietary incorporation of 15% *L. digitata* for broiler chickens impaired animal growth performance but improved meat nutritional value due to an increase in antioxidant carotenoids and chlorophylls, as well as a percentual increment in n-3 LC-PUFAs, such as 20:5n-3, 22:5n-3, and 22:6n-3, in muscle. The seaweed was also responsible for an increase in iodine in meat, which could be beneficial for human health. However, for longer trials, caution must be taken in terms of the maximum dose of iodine in feed to maintain animal welfare. Although dietary macroalgae decreased breast meat redness and reduced meat flavor intensity and juiciness, these results are not expected to influence consumers’ decisions toward meat, since no effect of treatments on meat visual perception and overall acceptability was detected by the trained sensory panel. The commercial (Rovabio^®^ Excel AP) and recombinant (alginate lyase) CAZymes reduced the negative impact on growth performance caused by the high levels of macroalgae in feed, while maintaining the positive effects of macroalgae on meat quality parameters. In addition, alginate lyase decreased intestinal viscosity, which could enhance the digestibility and absorption of nutrients. Therefore, it would be interesting to pursue studies to analyze the effect on nutrient digestibility of broilers fed *L. digitata* combined with the recombinant CAZyme or other novel exogenous enzymes. Feeding broiler chickens with lower levels of this macroalga would also be a promising nutritional strategy for reversing the negative influence of *L. digitata* on animal growth performance. Considering the economic and environmental impact of macroalga production, further studies performing a circular economy analysis are expected to provide a sustainability scenario of the incorporation of seaweed supplemented with feed enzymes in broiler’s diet.

## Figures and Tables

**Figure 1 animals-12-01007-f001:**
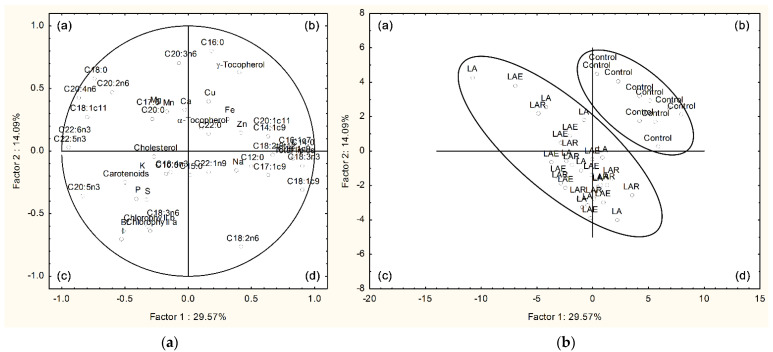
Principal component analysis of variables from breast meat of broilers fed *L. digitata*, individually or combined with exogenous CAZymes: (**a**) loading plot of the first and second principal components of pooled data; (**b**) component score vectors. Dietary treatments: control, corn–soybean basal diet; LA, basal diet plus 15% *L. digitata*; LAR, basal diet plus 15% *L. digitata* + 0.005% Rovabio^®^ Excel AP; LAE, basal diet plus 15% *L. digitata* + 0.01% recombinant CAZyme; (a–d) represent the quadrants a, b, c, and d.

**Table 1 animals-12-01007-t001:** Ingredients and feed additives of the experimental diets (percentage as fed basis).

	Dietary Treatments ^1^			
Item	Control	LA	LAR	LAE
Corn	50.4	32.6	32.6	32.6
Soybean meal	41.2	42.9	42.9	42.9
Sunflower oil	4.80	6.93	6.93	6.93
Sodium chloride	0.38	0.00	0.00	0.00
Calcium carbonate	1.10	0.90	0.90	0.90
Dicalcium phosphate	1.6	1.12	1.12	1.12
dl-Methionine	0.12	0.15	0.15	0.15
Vitamin–mineral premix ^2^	0.40	0.40	0.40	0.40
*Laminaria digitata* powder	-	15.0	15.0	15.0
Rovabio^®^ Excel AP	-	-	0.005	-
Recombinant CAZyme	-	-	-	0.01

^1^ Control, corn–soybean basal diet; LA, basal diet plus 15% *L. digitata*; LAR, basal diet plus 15% *L. digitata* + 0.005% Rovabio^®^ Excel AP; LAE, basal diet plus 15% *L. digitata* + 0.01% recombinant CAZyme. ^2^ Premix provided the following nutrients per kg of diet: pantothenic acid 10 mg, vitamin D_3_ 2400 IU, cyanocobalamin 0.02 mg, folic acid 1 mg, vitamin K_3_ 2 mg, nicotinic acid 25 mg, vitamin B_6_ 2 mg, vitamin A 10,000 UI, vitamin B_1_ 2 mg, vitamin E 30 mg, vitamin B_2_ 4 mg, Cu 8 mg, Fe 50 mg, I 0.7 mg, Mn 60 mg, Se 0.18 mg, and Zn 40 mg.

**Table 2 animals-12-01007-t002:** Chemical composition of *L. digitata* and experimental diets.

	Macroalga	Dietary Treatments ^1^			
Item	*L. digitata*	Control	LA	LAR	LAE
Energy (kcal ME/kg dry matter)	3065	4178	4184	4209	4201
Proximate composition (% as dry matter)					
Dry matter	90.8	89.8	89.8	90.1	90.1
Crude protein	4.85	23.0	23.7	23.5	23.3
Crude fat	1.31	8.28	9.95	10.10	10.20
Ash	17.4	6.5	7.4	7.5	7.6
Amino-acid composition (% as fed basis)					
Arginine	-	1.54	1.54	1.54	1.54
Histidine	-	0.59	0.57	0.57	0.57
Isoleucine	-	1.12	1.12	1.12	1.12
Leucine	-	1.91	1.80	1.80	1.80
Lysine	-	1.23	1.24	1.24	1.24
Methionine	-	0.47	0.48	0.48	0.48
Phenylalanine	-	1.22	1.19	1.19	1.19
Threonine	-	0.85	0.83	0.83	0.83
Tryptophan	-	0.32	0.32	0.32	0.32
Valine	-	1.20	1.17	1.17	1.17
Fatty-acid profile (% total fatty acids)					
14:0	5.12	0.088	0.206	0.207	0.214
16:0	22.7	9.13	8.78	8.84	8.79
16:1c9	2.95	0.114	0.174	0.175	0.175
17:0	0.454	0.051	0.050	0.049	0.047
17:1c9	0.581	0.026	0.036	0.038	0.039
18:0	1.09	3.05	3.10	3.13	3.17
18:1c9	19.3	27.5	26.5	26.4	27.2
18:2n-6	8.32	56.4	56.8	56.8	56.1
18:3n-3	5.14	0.888	0.935	0.932	0.919
18:4n-3	5.84	0.005	0.143	0.149	0.149
20:0	0.931	0.345	0.316	0.324	0.320
20:4n-6	9.79	0.001	0.208	0.211	0.218
20:5n-3	13.8	0.004	0.276	0.280	0.290
Diterpene profile (µg/g)					
α-Tocopherol	38.2	71.4	92.2	85.8	81.7
α-Tocotrienol	n.d.	6.39	3.55	3.19	3.09
γ-Tocopherol	0.180	1.07	1.10	0.929	0.960
γ-Tocopherol + β-tocotrienol	0.129	5.68	3.44	3.06	3.26
γ-Tocotrienol	n.d.	6.13	3.85	3.25	3.23
δ-Tocopherol	n.d.	1.11	0.759	0.597	0.730
Pigments (µg/g) ^2^					
β-Carotene	7.34	0.846	3.30	3.49	2.87
Chlorophyll a	235	1.72	58.6	57.2	58.6
Chlorophyll b	4.40	0.566	1.01	0.855	1.21
Total chlorophylls	239	2.29	59.6	58.01	59.8
Total carotenoids	93.9	2.87	21.6	21.6	21.7
Total chlorophylls + carotenoids	333	5.16	81.2	79.6	81.5
Mineral profile (mg/kg dry matter)					
Arsenic	40.3	n.d.	n.d.	n.d.	n.d.
Barium	5.94	n.d.	n.d.	n.d.	n.d.
Bromine	474	4.79	131	131	122
Cadmium	0.072	n.d.	n.d.	n.d.	n.d.
Calcium	8819	28,128	17,327	18,392	17,530
Chromium	2.13	n.d.	n.d.	n.d.	n.d.
Cobalt	n.d.	n.d.	n.d.	n.d.	n.d.
Copper	2.88	26.68	16.06	15.42	15.68
Iodine	4399	1.64	1068	1076	1036
Iron	144	407	237	274	241
Lead	n.d.	n.d.	n.d.	n.d.	n.d.
Magnesium	5637	2648	3326	3466	3276
Manganese	5.42	218	154	171	160
Nickel	n.d.	n.d.	n.d.	n.d.	n.d.
Phosphorus	903	12,129	7647	7881	7673
Potassium	28,530	15,676	19,011	19,237	18,596
Sodium	22,627	3563	5807	6495	6077
Sulfur	7653	4012	4474	4664	4599
Vanadium	1.34	n.d.	n.d.	n.d.	n.d.
Zinc	28.1	233	147	168	145

^1^ Control, corn–soybean basal diet; LA, basal diet plus 15% *L. digitata*; LAR, basal diet plus 15% *L. digitata* + 0.005% Rovabio^®^ Excel AP; LAE, basal diet plus 15% *L. digitata* + 0.01% recombinant CAZyme. ^2^ Pigments were determined using the equations described by Hynstova et al. (2018). n.d., not detected.

**Table 3 animals-12-01007-t003:** Growth performance (day 21–day 35), relative weight and length of gastrointestinal (GI) tract, and intestinal content viscosity of broilers (*n* = 10).

	Dietary Treatments ^1^				SEM ^2^	*p*-Value
Item	Control	LA	LAR	LAE		
Growth performance						
Initial body weight (g)	809.0	759.1	741.5	739.1	28.62	0.294
Final body weight (g)	1823 ^a^	1631 ^b^	1706 ^a,b^	1644 ^b^	42.4	0.011
ADG ^3^ (g/day)	78.9 ^a^	67.6 ^b^	74.8 ^a,b^	70.4 ^a,b^	2.82	0.039
ADFI ^4^ (g/pen)	130	126	125	127	3.1	0.523
Feed conversion ratio	1.70 ^b^	1.89 ^a^	1.82 ^a,b^	1.81 ^a,b^	0.038	0.012
Relative weight of GI tract (g/kg BW)						
Crop	2.99	3.66	3.43	3.92	0.143	0.125
Gizzard	17.3 ^a^	15.1 ^b^	14.9 ^b^	14.9 ^b^	0.32	0.013
Pancreas	2.32	2.52	2.39	2.44	0.057	0.664
Liver	24.0	22.4	22.9	22.9	0.31	0.285
Duodenum	5.58	6.47	6.56	6.20	0.164	0.139
Jejunum	11.1	12.6	12.9	11.9	0.30	0.164
Ileum	9.37 ^b^	11.5 ^a^	11.4 ^a^	10.6 ^a,b^	0.27	0.012
Caecum ^5^	5.12 ^c^	7.75 ^a^	7.25 ^a,b^	6.06 ^b,c^	0.279	0.001
Relative length of GI tract (cm/kg BW)						
Duodenum	17.4	19.2	18.5	19.2	0.33	0.190
Jejunum	41.1 ^b^	48.3 ^a^	46.0 ^a^	45.3 ^a^	0.80	0.010
Ileum	43.8 ^b^	56.3 ^a^	54.2 ^a^	54.2 ^a^	0.99	<0.001
Caecum ^4^	11.8 ^b^	14.6 ^a^	13.2 ^a,b^	14.7 ^a^	0.31	0.001
Content viscosity (cP)						
Duodenum + jejunum	4.28	5.10	5.17	4.60	0.153	0.118
Ileum	6.38 ^b^	10.92 ^a^	11.39 ^a^	6.84 ^b^	0.542	<0.001

^1^ Control, corn–soybean basal diet; LA, basal diet plus 15% *L. digitata*; LAR, basal diet plus 15% *L. digitata* + 0.005% Rovabio^®^ Excel AP; LAE, basal diet plus 15% *L. digitata* + 0.01% recombinant CAZyme. ^2^ SEM, standard error of the mean. ^3^ ADG, average daily gain. ^4^ ADFI, average daily feed intake. ^5^ Cecum: weight of two ceca. ^a,b,c^ Different superscript letters within a row indicate a significant difference (*p* < 0.05).

**Table 4 animals-12-01007-t004:** Meat quality and carcass traits of broilers (*n* = 10).

	Dietary Treatments ^1^				SEM ^2^	*p*-Value
Item	Control	LA	LAR	LAE		
Breast						
pH 24 h	5.75 ^a,b^	5.79 ^a^	5.76 ^a,b^	5.64 ^b^	0.034	0.020
Colour parameters						
Lightness (L*)	97.2	95.0	94.8	95.5	1.34	0.595
Redness (a*)	1.88 ^a^	−0.04 ^b^	0.73 ^a,b^	0.39 ^a,b^	0.453	0.032
Yellowness (b*)	4.10	4.51	3.29	3.94	0.638	0.605
Cooking loss (%)	27.6	26.1	26.1	25.4	1.00	0.498
Shear force (kg)	2.77	2.66	2.42	2.68	0.186	0.597
Thigh						
pH 24 h	5.91 ^b^	6.09 ^a^	6.03 ^a,b^	5.99 ^a,b^	0.042	0.035
Color parameters						
Lightness (L*)	48.6	47.2	46.9	47.6	0.54	0.134
Redness (a*)	8.57	8.85	9.67	8.77	0.446	0.332
Yellowness (b*)	6.05	5.58	5.36	5.53	0.348	0.551
Cooking loss (%)	24.5	26.1	26.4	25.3	0.74	0.275
Shear force (kg)	2.07	2.38	2.19	2.16	0.116	0.296

^1^ Control, corn–soybean basal diet; LA, basal diet plus 15% *L. digitata*; LAR, basal diet plus 15% *L. digitata* + 0.005% Rovabio^®^ Excel AP; LAE, basal diet plus 15% *L. digitata* + 0.01% recombinant CAZyme. ^2^ SEM, standard error of the mean. ^a,b^ Different superscript letters within a row indicate a significant difference (*p* < 0.05).

**Table 5 animals-12-01007-t005:** Sensorial attributes of broiler breast meat (*n* = 10).

	Dietary Treatments ^1^				SEM ^2^	*p*-Value
Item	Control	LA	LAR	LAE		
Tenderness	5.57	5.54	5.60	5.15	0.140	0.081
Juiciness	5.07 ^a^	4.74 ^a,b^	4.91 ^a^	4.35 ^b^	0.139	0.002
Flavor	5.85 ^a^	5.69 ^a,b^	5.70 ^a,b^	5.31 ^b^	0.113	0.007
Off-flavor	0.179	0.158	0.271	0.291	0.0839	0.604
Overall acceptability	5.13	5.17	5.17	4.71	0.155	0.099

^1^ Control, corn–soybean basal diet; LA, basal diet plus 15% *L. digitata*; LAR, basal diet plus 15% *L. digitata* + 0.005% Rovabio^®^ Excel AP; LAE, basal diet plus 15% *L. digitata* + 0.01% recombinant CAZyme. ^2^ SEM, standard error of the mean. ^a,b^ Different superscript letters within a row indicate a significant difference (*p* < 0.05).

**Table 6 animals-12-01007-t006:** Diterpene profile and pigments in breast and thigh meats of broilers (*n* = 10).

	Dietary Treatments ^1^				SEM ^2^	*p*-Value
Item	Control	LA	LAR	LAE		
Breast						
Malondialdehyde (mg/kg)						
day 0	-	-	-	-	-	-
day 6	-	-	-	-	-	-
Diterpene profile (µg/g)						
α-Tocopherol	5.14	5.12	4.68	4.52	0.330	0.448
γ-Tocopherol	0.072 ^a^	0.055 ^b^	0.048 ^b^	0.052 ^b^	0.0030	<0.001
Pigments (µg/100 g) ^3^						
Chlorophyll a	16.2 ^b^	32.6 ^a^	29.8 ^a^	32.6 ^a^	2.66	<0.001
Chlorophyll b	28.2 ^b^	48.0 ^a^	48.0 ^a^	55.5 ^a^	3.73	<0.001
Total chlorophylls	44.3 ^b^	80.7 ^a^	81.2 ^a^	88.1 ^a^	6.25	<0.001
Total carotenoids	36.9 ^b^	49.2 ^a^	54.1 ^a^	48.5 ^a^	2.97	0.002
Total chlorophylls + carotenoids	81.3 ^b^	130 ^a^	135 ^a^	137 ^a^	6.63	<0.001
Thigh						
Malondialdehyde (mg/kg)						
day 0	n.d.	n.d.	n.d.	n.d.	-	-
day 6	0.702	0.405	0.480	0.153	0.154	0.111
Diterpene profile (µg/g)						
α-Tocopherol	7.60	7.47	6.56	6.38	0.404	0.085
γ-Tocopherol	0.075 ^a^	0.068 ^a,b^	0.062 ^a,b^	0.060 ^b^	0.0038	0.030
Pigments (µg/100 g) ^2^						
Chlorophyll a	17.4 ^b^	28.0 ^a^	29.4 ^a^	27.5 ^a^	1.61	<0.001
Chlorophyll b	29.5 ^b^	46.3 ^a^	50.7 ^a^	44.8 ^a^	3.31	<0.001
Total chlorophylls	46.9 ^b^	74.3 ^a^	80.1 ^a^	72.3 ^a^	4.85	<0.001
Total carotenoids	35.5 ^b^	49.2 ^a^	49.2 ^a^	54.2 ^a^	3.21	0.002
Total chlorophylls + carotenoids	82.4 ^b^	123 ^a^	129 ^a^	126 ^a^	5.25	<0.001

^1^ Control, corn–soybean basal diet; LA, basal diet plus 15% *L. digitata*; LAR, basal diet plus 15% *L. digitata* + 0.005% Rovabio^®^ Excel AP; LAE, basal diet plus 15% *L. digitata* + 0.01% recombinant CAZyme. ^2^ SEM, standard error of the mean. ^3^ Pigments were determined using the equations described by Hynstova et al. [29]. ^a,b^ Different superscript letters within a row indicate a significant difference (*p* < 0.05).

**Table 7 animals-12-01007-t007:** Total lipid content, cholesterol content, and fatty-acid (FA) composition in the breast meat of broilers (*n* = 10).

	Dietary Treatments ^1^				SEM ^2^	*p*-Value
Item	Control	LA	LAR	LAE		
Total lipids (g/100 g)	1.69 ^a^	1.25 ^b^	1.25 ^b^	1.23 ^b^	0.093	0.003
Cholesterol (mg/g)	0.463	0.561	0.569	0.547	0.0443	0.318
FA composition (g/100 g FA)						
10:0	0.001	0.002	0.006	0.004	0.0020	0.441
12:0	0.036	0.037	0.036	0.029	0.0028	0.208
14:0	0.29 ^a^	0.23 ^b^	0.25 ^a,b^	0.22 ^b^	0.011	0.001
14:1c9	0.023 ^a^	0.007 ^b^	0.006 ^b^	0.002 ^b^	0.0043	0.006
15:0	0.071	0.082	0.080	0.072	0.0044	0.236
16:0	15.3 ^a^	13.9 ^b^	13.9 ^b^	13.7 ^b^	0.24	<0.001
16:1c7	0.34 ^a^	0.26 ^b^	0.28 ^b^	0.26 ^b^	0.014	<0.001
16:1c9	0.92 ^a^	0.67 ^b^	0.69 ^a,b^	0.66 ^b^	0.062	0.014
17:0	0.19	0.20	0.19	0.19	0.011	0.874
17:1c9	0.039	0.024	0.035	0.024	0.0048	0.083
18:0	8.57	9.14	8.69	9.18	0.394	0.609
18:1c9	25.6	23.1	24.5	23.6	0.65	0.052
18:1c11	1.21	1.42	1.42	1.39	0.059	0.049
18:2n-6	36.7	37.8	38.4	37.6	0.88	0.575
18:3n-6	0.14 ^b^	0.15 ^a,b^	0.16 ^a^	0.16 ^a^	0.004	0.001
18:2t9t12	0.29 ^a^	0.25 ^a,b^	0.24 ^b^	0.24 ^b^	0.015	0.027
18:3n-3	0.55 ^a^	0.41 ^b^	0.40 ^b^	0.42 ^b^	0.025	<0.001
18:4n-3	0.004	0.015	0.068	0.004	0.0246	0.218
20:0	0.094	0.090	0.094	0.090	0.0032	0.670
20:1c11	0.23 ^a^	0.19 ^b^	0.20 ^b^	0.20 ^b^	0.007	0.001
20:2n-6	0.56	0.62	0.56	0.59	0.036	0.637
20:3n-6	0.63 ^a^	0.56 ^a,b^	0.48 ^b^	0.55 ^a,b^	0.029	0.012
20:4n-6	4.59	6.01	6.08	5.16	0.507	0.132
20:3n-3	n.d.	n.d.	n.d.	n.d.	n.d.	n.d.
20:5n-3	0.025 ^b^	0.097 ^a^	0.108 ^a^	0.095 ^a^	0.0076	<0.001
22:0	0.039	0.027	0.040	0.037	0.0077	0.615
22:1n-9	0.004	0.006	0.024	0.000	0.0095	0.308
22:2n-6	n.d.	n.d.	n.d.	n.d.	n.d.	n.d.
22:5n-3	0.21 ^b^	0.58 ^a^	0.46 ^a^	0.57 ^a^	0.049	<0.001
22:6n-3	0.12 ^b^	0.47 ^a^	0.32 ^a,b^	0.49 ^a^	0.065	0.001
Others	3.23	3.60	3.23	3.46	0.332	0.820
Partial sums of FA, g/100 g FA						
SFA	24.6	23.8	23.2	23.6	0.58	0.433
*cis*-MUFA	28.4 ^a^	25.8 ^b^	27.1 ^a,b^	26.2 ^a,b^	0.68	0.045
PUFA	43.8 ^b^	46.9 ^a^	46.4 ^a^	46.8 ^a^	0.48	<0.001
n-6 PUFA	42.6 ^b^	45.1 ^a^	44.8 ^a^	45.0 ^a^	0.50	0.003
n-3 PUFA	0.90 ^b^	1.57 ^a^	1.38 ^a^	1.57 ^a^	0.101	<0.001
Ratios of FA						
PUFA/SFA	1.79 ^b^	2.00 ^a^	2.01 ^a^	2.00 ^a^	0.062	0.046
n-6/n-3	47.7 ^a^	30.2 ^b^	34.4 ^b^	29.6^b^		<0.001

^1^ Control, corn–soybean basal diet; LA, basal diet plus 15% *L. digitata*; LAR, basal diet plus 15% *L. digitata* + 0.005% Rovabio^®^ Excel AP; LAE, basal diet plus 15% *L. digitata* + 0.01% recombinant CAZyme. ^2^ SEM, standard error of the mean. SFA = sum of (10:0, 12:0, 14:0, 15:0, 16:0, 17:0, 18:0, 20:0, 22:0). *cis*-MUFA = sum of (14:1c9, 16:1c7, 16:1c9, 17:1c9, 18:1c9, 18:1c11, 20:1c11, 22:1n-9). PUFA = sum of (18:2n-6, 18:2t9t12, 18:3n-6, 18:3n-3, 18:4n-3, 20:2n-6, 20:3n-6, 20:4n-6, 20:3n-3, 20:5n-3, 22:5n-3, 22:6n-3). n-6 PUFA = sum of (18:2n-6, 18:3n-6, 20:2n-6, 20:3n-6, 20:4n-6). n-3 PUFA = sum of (18:3n-3, 18:4n-3, 20:3n-3, 20:5n-3, 22:5n-3, 22:6n-3). ^a,b^ Different superscript letters within a row indicate a significant difference (*p* < 0.05). n.d., not detected.

**Table 8 animals-12-01007-t008:** Total lipid content, cholesterol content, and fatty-acid (FA) composition in the thigh meat of broilers (*n* = 10).

	Dietary Treatments ^1^				SEM ^2^	*p*-Value
Item	Control	LA	LAR	LAE		
Total lipids (g/100 g)	3.35 ^a^	2.36 ^b^	2.56 ^b^	2.22 ^b^	0.134	<0.001
Cholesterol (mg/g)	0.664	0.611	0.657	0.571	0.0376	0.280
FA composition (g/100 g FA)						
10:0	0.004	0.009	0.009	0.005	0.0020	0.103
12:0	0.042	0.038	0.044	0.038	0.0023	0.159
14:0	0.33 ^a^	0.27 ^b^	0.29 ^b^	0.28 ^b^	0.010	0.002
14:1c9	0.035 ^a^	0.024 ^b^	0.025 ^b^	0.029 ^a,b^	0.0024	0.014
15:0	0.077	0.074	0.081	0.083	0.0026	0.092
16:0	15.6 ^a^	13.6 ^b^	14.2 ^b^	13.5 ^b^	0.27	<0.001
16:1c7	0.41 ^a^	0.32 ^b^	0.34 ^b^	0.36 ^a,b^	0.013	<0.001
16:1c9	1.43 ^a^	1.10 ^b^	1.15 ^a,b^	1.19 ^a,b^	0.081	0.033
17:0	0.17	0.17	0.18	0.17	0.006	0.850
17:1c9	0.045	0.036	0.040	0.039	0.0021	0.093
18:0	7.38	8.02	7.73	7.40	0.315	0.437
18:1c9	29.3	27.4	28.9	28.5	0.51	0.063
18:1c11	1.12 ^b^	1.33 ^a^	1.30 ^a^	1.26 ^a^	0.037	0.001
18:2n-6	37.5	39.9	38.6	39.9	0.82	0.134
18:3n-6	0.10	0.11	0.11	0.11	0.004	0.755
18:2t9t12	0.28 ^a^	0.21 ^b^	0.18 ^b^	0.22 ^a,b^	0.017	0.003
18:3n-3	0.58 ^a^	0.45 ^b^	0.45 ^b^	0.46 ^b^	0.027	0.002
18:4n-3	0.017	0.020	0.021	0.022	0.0034	0.770
20:0	0.09	0.10	0.10	0.10	0.005	0.589
20:1c11	0.23	0.22	0.23	0.21	0.006	0.060
20:2n-6	0.36	0.38	0.35	0.35	0.013	0.294
20:3n-6	0.42 ^a^	0.34 ^b^	0.31 ^b^	0.32 ^b^	0.018	0.001
20:4n-6	2.14 ^b^	3.04 ^a^	2.55 ^a,b^	2.70 ^a,b^	0.168	0.006
20:3n-3	0.014 ^a^	0.008 ^a,b^	0.001 ^b^	0.002 ^a,b^	0.0032	0.032
20:5n-3	0.013 ^b^	0.046 ^a^	0.036 ^a^	0.041 ^a^	0.0037	<0.001
22:0	0.040	0.046	0.047	0.046	0.0024	0.176
22:1n-9	0.058	0.062	0.061	0.048	0.0147	0.889
22:2n-6	0.008	0.009	0.006	0.008	0.0019	0.726
22:5n-3	0.10 ^b^	0.29 ^a^	0.21 ^a^	0.23 ^a^	0.019	<0.001
22:6n-3	0.09 ^b^	0.30 ^a^	0.23 ^a^	0.26 ^a^	0.026	<0.001
Others	2.07	2.17	2.16	2.09	0.104	0.879
Partial sums of FA, g/100 g FA						
SFA	23.7	22.3	22.7	21.7	0.51	0.052
*cis*-MUFA	31.2	29.4	30.9	30.4	0.53	0.095
PUFA	41.6	45.1	43.1	44.7	0.90	0.041
n-6 PUFA	40.5	43.7	41.9	43.4	0.87	0.053
n-3 PUFA	0.81 ^c^	1.11 ^a^	0.95 ^b,c^	1.02 ^a,b^	0.037	<0.001
Ratios of FA						
PUFA/SFA	1.76 ^b^	2.04 ^a,b^	1.92 ^a,b^	2.07 ^a^	0.079	0.040
n-6/n-3	50.1 ^a^	40.0 ^c^	44.4 ^b^	42.7 ^b,c^	1.10	<0.001

^1^ Control, corn–soybean basal diet; LA, basal diet plus 15% *L. digitata*; LAR, basal diet plus 15% *L. digitata* + 0.005% Rovabio^®^ Excel AP; LAE, basal diet plus 15% *L. digitata* + 0.01% recombinant CAZyme. ^2^ SEM, standard error of the mean. SFA = sum of (10:0, 12:0, 14:0, 15:0, 16:0, 17:0, 18:0, 20:0, 22:0). *cis*-MUFA = sum of (14:1c9, 16:1c7, 16:1c9, 17:1c9, 18:1c9, 18:1c11, 20:1c11, 22:1n-9). PUFA = sum of (18:2n-6, 18:2t9t12, 18:3n-6, 18:3n-3, 18:4n-3, 20:2n-6, 20:3n-6, 20:4n-6, 20:3n-3, 20:5n-3, 22:5n-3, 22:6n-3). n-6 PUFA = sum of (18:2n-6, 18:3n-6, 20:2n-6, 20:3n-6, 20:4n-6). n-3 PUFA = sum of (18:3n-3, 18:4n-3, 20:3n-3, 20:5n-3, 22:5n-3, 22:6n-3). ^a,b,c^ Different superscript letters within a row indicate a significant difference (*p* < 0.05). n.d., not detected.

**Table 9 animals-12-01007-t009:** Mineral composition of breast and thigh meats of broilers (*n* = 10).

	Dietary Treatments ^1^				SEM ^2^	*p*-Value
Item	Control	LA	LAR	LAE		
Breast						
Macrominerals (mg/100 g fresh weight)						
Calcium	17.0	17.2	16.9	16.9	0.86	0.993
Magnesium	32.7	32.3	32.4	34.2	1.24	0.695
Phosphorus	242	255	254	256	5.2	0.225
Potassium	499	518	506	508	8.7	0.494
Sodium	62.3	64.1	62.6	60.0	2.23	0.628
Sulfur	211	226	223	223	5.2	0.209
Total	1064	1112	1096	1098	16.4	0.226
Microminerals (mg/100 g fresh weight)						
Bromine	0.12 ^b^	0.85 ^a^	0.78 ^a^	0.74 ^a^	0.039	<0.001
Copper	0.078	0.070	0.075	0.074	0.0029	0.270
Iodine	0.01 ^b^	0.36 ^a^	0.35 ^a^	0.35 ^a^	0.022	<0.001
Iron	1.02	0.88	0.96	0.96	0.037	0.083
Manganese	0.065	0.068	0.068	0.065	0.0027	0.787
Zinc	1.32	1.23	1.23	1.17	0.052	0.290
Total	2.61 ^b^	3.46 ^a^	3.46 ^a^	3.35 ^a^	0.112	<0.001
Total macro- and microminerals	1067	1116	1099	1101	16.4	0.211
Thigh						
Macrominerals (mg/100 g fresh weight)						
Calcium	17.8	16.7	17.7	17.0	0.39	0.125
Magnesium	29.7	29.9	29.4	29.8	0.58	0.949
Phosphorus	225	231	224	232	2.8	0.126
Potassium	470	478	462	475	5.0	0.149
Sodium	63.0	62.3	62.9	61.4	1.57	0.880
Sulfur	202	211	204	208	3.7	0.333
Total	1008	1029	1001	1023	10.0	0.170
Microminerals (mg/100 g fresh weight)						
Bromine	-	-	-	-	-	-
Copper	0.099	0.093	0.098	0.098	0.0038	0.646
Iodine	-	-	-	-	-	-
Iron	0.780	0.792	0.831	0.844	0.0346	0.510
Manganese	0.064	0.063	0.063	0.066	0.0020	0.604
Zinc	2.38	2.35	2.51	2.36	0.077	0.434
Total	3.32	3.30	3.50	3.37	0.089	0.383
Total macro- + microminerals	1011	1032	1004	1026	10.0	0.173

^1^ Control, corn–soybean basal diet; LA, basal diet plus 15% *L. digitata*; LAR, basal diet plus 15% *L. digitata* + 0.005% Rovabio^®^ Excel AP; LAE, basal diet plus 15% *L. digitata* + 0.01% recombinant CAZyme. ^2^ SEM, standard error of the mean. ^a,b^ Different superscript letters within a row indicate a significant difference (*p* < 0.05).

**Table 10 animals-12-01007-t010:** Loadings for the first two principal components in the breast muscle.

Variables	Factor 1	Factor 2
10:0	−0.18	−0.18
12:0	0.50	−0.12
14:0	0.90	0.00
14:1c9	0.63	0.12
15:0	0.01	−0.19
16:0	0.18	0.80
16:1c7	0.83	0.02
16:1c9	0.83	−0.05
17:0	−0.33	0.32
17:1c9	0.63	−0.19
18:0	−0.74	0.58
18:1c9	0.90	−0.31
18:1c11	−0.80	0.28
18:2n-6	0.42	−0.77
18:3n-6	−0.21	−0.56
18:2t9t12	0.67	−0.03
18:3n-3	0.90	−0.11
18:4n-3	−0.14	−0.17
20:0	−0.29	0.26
20:1c11	0.65	0.17
20:2n-6	−0.60	0.47
20:3n-6	−0.07	0.71
20:4n-6	−0.87	0.43
20:5n-3	−0.84	−0.36
22:0	0.16	0.14
22:1n-9	0.16	−0.17
22:5n-3	−0.95	0.03
22:6n-3	−0.91	0.10
Total lipids	0.83	−0.06
Cholesterol	−0.27	−0.04
α-Tocopherol	0.10	0.19
γ-Tocopherol	0.40	0.63
Chlorophyll a	−0.30	−0.64
Chlorophyll b	−0.32	−0.60
Carotenoids	−0.50	−0.25
Sodium	0.38	−0.15
Potassium	−0.37	−0.19
Calcium	−0.03	0.33
Magnesium	−0.27	0.34
Phosphorus	−0.41	−0.38
Sulfur	−0.33	−0.39
Copper	0.16	0.40
Zinc	0.42	0.15
Manganese	−0.16	0.32
Iron	0.32	0.26
Bromine	−0.51	−0.64
Iodine	−0.53	−0.71

## Data Availability

The data presented in this study are available in this article (and Appendix A).

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
