# Peer review of "Effect of Dietary Laminaria digitata with Carbohydrases on Broiler Production Performance and Meat Quality, Lipid Profile, and Mineral Composition"

_animals, 2022, doi:10.3390/ani12081007_

Round 1
Reviewer 1 Report
Dear authors, this study provides new and important data about the effects of the supplementation in broiler diets of Laminaria digitata with carbohydrases enzyme addition or not on broiler performance parameters and meat quality. Moreover, the use of English language in the present study was appropriate and only some phrases must be rewritten. However, authors must answer several questions according to mainly to the zootechnical and secondly to the statistical methods used. Moreover, more studies investigating the dietary supplementation of Laminaria digitata in broiler and pig diets and their effects on nutritional and zootechnical parameters must be added in the manuscript. Due to this fact, authors must do better research and add the appropriate data and references to the manuscript. Secondly, despite that the present study had the appropriate treatments only n=120 birds were used and were not 1d old in order to exert more certain results but 22d old. Thirdly, the discussion section is too long and several paragraphs must be shortened or deleted. Conclusively, this study, needs to be major revised in order to be published in this journal. The questions which must be answered and the changes which are proposed to be done, are presented line by line in the following paragraphs.
Simple Summary:
L.27-28-29. Have you done any economic evaluation in order to decide if these supplementations of Laminaria digitata and carbohydrases on broiler diets are cost efficient?
Abstract:
L.38. How did you decide to use these concentrations of Laminaria digitata supplementation in broiler diets?
L.40: Low number of broilers was used in the present experiment (only 120). Why this experiment was not conducted in 1d old broilers in order to exert more certain results?
Introduction:
L.71-72. “To the best of our knowledge, despite the potential of using Laminaria sp. as a feed 71 ingredient, such application for monogastric animals was scarcely reported”. With a little research 2 studies evaluating the effects of Laminaria digitata supplementation in broiler diets were found. Please add more references in your manuscript as for example:
Torres Sweeney, Hazel Meredith , Stafford Vigorsa , Mary J. McDonnell , Marion Ryan , Kevin Thornton, John V. O’Doherty. 2017. Extracts of laminarin and laminarin/fucoidan from the marine macroalgal species Laminaria digitata improved growth rate and intestinal structure in young chicks, but does not influence Campylobacter jejuni colonization. Animal Feed Science and Technology 232:71-79
B. Venardou J. V. O’Doherty,y S. Vigors, C. J. O’Shea , E. J. Burton, M. T. Ryan, and T. Sweeney. 2021. Effects of dietary supplementation with a laminarin-rich extract on the growth performance and gastrointestinal health in broilers. Poultry Science 100:101179
Materials and Methods:
L.117-118. Why authors evaluated the effects of the examined parameters only in 120 broilers and why the experiment was not conducted on 1d old broilers instead of 22d old day broilers which were used?
Statistical analysis:
Did authors examine data with non parametric tests?
Results:
----------------------------------------------------------
Discussion:
L. 534-537. This paragraph explained nothing please delete.
L. 538-595. Please shorten this paragraph.
Add more references that investigated the effects of Laminaria Digitata on broiler or pig nutrition. There are many examples in the international bibliography.
Conclusions:
The authors did not do an economic evaluation of the supplementation of Laminaria Digitata with carbohydrases in broiler nutrition. Please explain or refer to the conclusion.
Author Response
- Dear authors, this study provides new and important data about the effects of the supplementation in broiler diets of Laminaria digitata with carbohydrases enzyme addition or not on broiler performance parameters and meat quality. Moreover, the use of English language in the present study was appropriate and only some phrases must be rewritten. However, authors must answer several questions according to mainly to the zootechnical and secondly to the statistical methods used. Moreover, more studies investigating the dietary supplementation of Laminaria digitata in broiler and pig diets and their effects on nutritional and zootechnical parameters must be added in the manuscript. Due to this fact, authors must do better research and add the appropriate data and references to the manuscript. Secondly, despite that the present study had the appropriate treatments only n=120 birds were used and were not 1d old in order to exert more certain results but 22d old. Thirdly, the discussion section is too long and several paragraphs must be shortened or deleted. Conclusively, this study, needs to be major revised in order to be published in this journal. The questions which must be answered and the changes which are proposed to be done, are presented line by line in the following paragraphs.
Reply: thank you for your comments; we tried to address all of them, as described below.
- Simple Summary:
- 27-28-29. Have you done any economic evaluation in order to decide if these supplementations of Laminaria digitata and carbohydrases on broiler diets are cost efficient?
Reply: We did not any economic evaluation about the addition of Laminaria and carbohydrases on broiler diets and, therefore, no estimation on the economic feasibility of replacing 15% of corn grain with Laminaria digitata was performed. The use of high concentration of seaweed in the broiler diets is indeed expensive, despite a decrease of macroalgae cost is expectable. However, seaweeds can be sustainable alternatives to cereals as they do not compete with feed, food and fuel production and, although their cultivation technology is still in development in order to reduce production costs and ecological footprint, there are already sustainable seaweed production methods (i.e., Integrated Multitrophic Aquaculture, ITMA). Following steps should be the investment in Research and Development work in order to create feed, based on seaweeds, able to be produced at an industrial level and incorporated in broiler´s diets as alternative ingredients to the conventional feedstuffs.
- Abstract:
2.2. L.38. How did you decide to use these concentrations of Laminaria digitata supplementation in broiler diets?
Reply: The purpose of our study was to use Laminaria digitata as a feed ingredient (15% of dietary incorporation) in broiler diets and not as a supplement. To the best of our knowledge, there are fewer studies using macroalgae as an ingredient than as a supplement. However, according to Ventura et al. (1994) (https://doi.org/10.1016/0377-8401(94)90083-3), the maximum level of macroalgae in broiler diet was suggested to be up to 10% feed. However, the seaweed used in the latter study was Ulva rigida instead of L. digitata and, thus, we did not know what growth and meat quality effects would result from the use of L. digitata as an ingredient. So, we decided to go a little bit further on the dose and test the incorporation of a superior level of seaweed in the diet.
2.3. L.40: Low number of broilers was used in the present experiment (only 120). Why this experiment was not conducted in 1d old broilers in order to exert more certain results?
Reply: The number of broilers per experimental group allows the best possible cost-benefit combination, allowing a good statistical power and avoiding the use of an unnecessarily large number of animals, respecting the “Reduce” from the 3R’s principle (replacement, reduction and refinement), as stated in the EU Directive 2010/63/UE, which allows to minimize animal distress and maintain animal welfare. The experimental design has been previously approved by our Animal Ethics Commission and the National Authority. In the reports submitted to those entities the appropriate number of animals have been estimated based on the Power tool of SAS (n=10 is an acceptable value for the usual variability of growth and meat parameters analyzed). In addition, the same sample size (ref. 1, 2, 3) was used in previous studies by our team:
Pestana et al. 2020, https://doi.org/10.1016/j.psj.2019.11.069
Alfaia et al. 2021, https://doi.org/10.1016/j.psj.2020.11.034
Coelho et al. 2020, https://doi.org/10.3390/ani10122384
The experiment was not conducted in 1d old broilers because we aimed to test the effects of dietary treatments during the finishing period of broilers, from days 21 to 35, where 35 day-old is the standard slaughter age. Therefore, the experimental period (14 days corresponded to this stage of broiler growth. Before that, birds were fed ad libitum with maize-based diet, which corresponded to the adaptation period before trial when the animals are still developing their digestive systems.
The information related to sample size and experimental period is now added between lines 124 to 128, page 3, and lines 132 and 133, page 9.
- Introduction:
- 71-72. “To the best of our knowledge, despite the potential of using Laminaria sp. as a feed 71 ingredient, such application for monogastric animals was scarcely reported”. With a little research 2 studies evaluating the effects of Laminaria digitata supplementation in broiler diets were found. Please add more references in your manuscript as for example:
Torres Sweeney, Hazel Meredith, Stafford Vigorsa, Mary J. McDonnell, Marion Ryan, Kevin Thornton, John V. O’Doherty. 2017. Extracts of laminarin and laminarin/fucoidan from the marine macroalgal species Laminaria digitata improved growth rate and intestinal structure in young chicks, but does not influence Campylobacter jejuni colonization. Animal Feed Science and Technology 232:71-79
- Venardou J. V. O’Doherty,y S. Vigors, C. J. O’Shea , E. J. Burton, M. T. Ryan, and T. Sweeney. 2021. Effects of dietary supplementation with a laminarin-rich extract on the growth performance and gastrointestinal health in broilers. Poultry Science 100:101179
Reply: Thank you for your suggestions. The authors acknowledge the reviewer’s comment and citations for two indicated studies were added in lines 71 to 72, page 2.
- Materials and Methods:
4.1. L.117-118. Why authors evaluated the effects of the examined parameters only in 120 broilers and why the experiment was not conducted on 1d old broilers instead of 22d old day broilers which were used?
Reply: Thank you for your comment. This comment was already replied above (2.3)
- Statistical analysis:
- Did authors examine data with non-parametric tests?
Reply: We did not use non-parametric tests to evaluate data because our data follows parametric test assumptions (normal distribution and homogeneity of variances). In theses circumstances, the parametric tests (One-way ANOVA) are more powerful than non-parametric tests.
- Discussion:
6.1. L. 534-537. This paragraph explained nothing please delete.
Reply: The reviewer is right. The paragraph was deleted.
6.2. L. 538-595. Please shorten this paragraph.
Reply: The paragraph between lines 540 and 594 was shorten, as suggested.
6.3. Add more references that investigated the effects of Laminaria digitata on broiler or pig nutrition. There are many examples in the international bibliography.
Reply: The following references were added in line 554, page 21:
Torres Sweeney, Hazel Meredith, Stafford Vigorsa, Mary J. McDonnell, Marion Ryan, Kevin Thornton, John V. O’Doherty. 2017. https://doi.org/10.1016/j.anifeedsci.2017.08.001
- Venardou J. V. O’Doherty,y S. Vigors, C. J. O’Shea , E. J. Burton, M. T. Ryan, and T. Sweeney. 2021. https://doi.org/10.1016/j.psj.2021.101179
O'Doherty, Dillon, Figat, Callan, & Sweeney (2010). https://doi.org/10.1016/j.anifeedsci.2010.03.004
Draper, Walsh, McDonnell, & O'Doherty (2016). https://doi.org/10.2527/jas.2015-9776
Bouwhuis, M. A., Sweeney, T., Mukhopadhya, A., McDonnell, M. J., & O'Doherty, J. V. (2017). https://doi.org/10.1016/j.anifeedsci.2016.11.007
- Conclusions:
- The authors did not do an economic evaluation of the supplementation of Laminaria digitata with carbohydrases in broiler nutrition. Please explain or refer to the conclusion.
Reply: Thank you for your comment. The aim of the present study was not an economic evaluation of using dietary Laminaria digitata with carbohydrases. However, we are aware of the high costs of seaweed production and, thus, it would be interesting to perform a circular economy study. This aspect was added between lines 749 and 752, page 24.
Reviewer 2 Report
Dear authors,
I read your manuscript (animals-1657268) with a lot of interest and found much merit in your work. Overall, it is well written and fits the scope of the journal. The methods are appropriate and performed correctly. The results section is in a proper form and the discussion is well done. I really liked the paper.
Just some minor comments, that you can explain or correct easily.
Materials and Methods section
Line 128. Remove “at” between (Adisseo; Antony, France) and (LAR)
Line 189. The formula needs a reference
Results section
Table 3. It seems strange that the initial body weight of the Control group chickens is 70 g higher than the rest, but no differences are found. The non-significant differences are probably due to a small sample size or a large variability in weights. Can it be a conditioning factor of the final body weight? Please clarify this in the discussion section.
In the same table, you need to correct some superscripts that appear as regular numbers and letters.
Table 6. You didn't analyze malondaldehyde on the breast? If yes, express results as n.d. as in the thigh.
Table 9. Has there been any problem with the analysis of Bromine and Iodine in the thigh? Results of those analysis don’t appear in the table.
Discussion section
Line 534 to 537. These are the guidelines of the journal. Please remove the paragraph.
Line 624. The amount of chlorophyll in meat is noticeably lower than in plants: you found 80.1 micrograms/100 g, but in green part of the plants the content is up to 200,000 micrograms/100 g. So, you need to pin down the real impact of meat chlorophyll on a human diet.
Author Response
- Dear authors,
I read your manuscript (animals-1657268) with a lot of interest and found much merit in your work. Overall, it is well written and fits the scope of the journal. The methods are appropriate and performed correctly. The results section is in a proper form and the discussion is well done. I really liked the paper.
Reply: Thank you for your comment.
Just some minor comments that you can explain or correct easily.
- Materials and Methods section
- Line 128. Remove “at” between (Adisseo; Antony, France) and (LAR)
Reply: Done.
- Line 189. The formula needs a reference
Reply: The citation was added in line 194, page 5, and the corresponding reference between lines 841 and 842, page 29.
- Results section
- Table 3. It seems strange that the initial body weight of the Control group chickens is 70 g higher than the rest, but no differences are found. The non-significant differences are probably due to a small sample size or a large variability in weights. Can it be a conditioning factor of the final body weight? Please clarify this in the discussion section.
Reply: We understand the reviewer´s concern. The absence of differences for the initial body weight (d 21) was due to a large variability in weights (SEM = 28.62), although animals were maintained in the same conditions and the body weight of broilers at d 0 was uniformed to obtain minimal differences between groups: Control (44.8 g), LA (45.6 g), LAR (45.3 g), LAE (44.7 g), as well as low variability (SEM = 0.73) (see lines 188 to 189). However, the initial body weight was not considered a conditioning factor of final body weight because no significant differences were found for this parameter and the factor that clearly influenced final body weight was dietary algae incorporation. For instance, the body weight at d 21 numerically differed 49.9 g between control and LA treatment, whereas, at d 35, this difference was almost 4 times higher than that found at d 21 (192 g). These aspects are now added between lines 546 and 551, pages 20 to 21.
- In the same table, you need to correct some superscripts that appear as regular numbers and letters.
Reply: The superscripts were corrected in Table 3, as suggested.
- Table 6. You didn't analyse malondaldehyde on the breast? If yes, express results as n.d. as in the thigh.
Reply: We did not analyse malondaldehyde on the breast and that is why the corresponding values are not expressed as n.d. in Table 6 as in the thigh. The thigh muscle was selected for determining lipid oxidative stability instead of breast muscle, since this muscle is more prone to oxidation (see the explanation between lines 159 and 165).
- Table 9. Has there been any problem with the analysis of Bromine and Iodine in the thigh? Results of those analysis don’t appear in the table.
Reply: The results of bromine and iodine analyses in the thigh don’t appear in Table 9, since the analyses were not performed because they were too expensive. Therefore, we had to select only one muscle (breast or thigh) to evaluate these minerals and we selected the one with higher representativeness (percentage of carcass weight), which corresponded to the breast muscle (please, see the explanation between lines 157 and 158, page 4).
- Discussion section
- Line 534 to 537. These are the guidelines of the journal. Please remove the paragraph.
Reply: Done.
- Line 624. The amount of chlorophyll in meat is noticeably lower than in plants: you found 80.1 micrograms/100 g, but in green part of the plants the content is up to 200,000 micrograms/100 g. So, you need to pin down the real impact of meat chlorophyll on a human diet.
Reply: The real impact of meat chlorophyll on a human diet was pin down, as suggested by the reviewer. The sentence between lines 648 and 650, page 22, was removed. Indeed, the amount of chlorophylls deposited in the meat was about 74 times lower than the average present in diets containing macroalgae. However, we still considered that the significant increase of chlorophyll with algae treatments can contribute to enhance meat nutritional value. The bioavailability of chlorophylls from meat is scarcely known and strongly depends on their deposition form, since their effective absorption on the human organism requires a complex mechanism of digestion.
Reviewer 3 Report
The work deals with a very interesting issue that has been treated very broadly. I congratulate the Authors on their diligence in the development of the manuscript and a very thorough discussion of the results obtained. I noted only very minor editorial errors in the literature list, e.g. in item 49 the name of the journal was not given, while in tab. 3 the letters of significance of differences should be given in superscript.
Author Response
- The work deals with a very interesting issue that has been treated very broadly. I congratulate the Authors on their diligence in the development of the manuscript and a very thorough discussion of the results obtained.
Reply: Thank you for your comment.
- I noted only very minor editorial errors in the literature list, e.g. in item 49 the name of the journal was not given, while in tab. 3 the letters of significance of differences should be given in superscript.
Reply: The name of the journal for the reference (now 56) is present between lines 921 and 922, page 29. The letters of significance in Table 3 are now given in superscript.
Round 2
Reviewer 1 Report
Dear authors, all the appropriate changes have been done, and all the questions have been answered properly. I recommend this paper, be published in Animals Journal.